# Simultaneous Inversion of Shallow Seismic Data for Imaging of Sulfurized Carbonates

**Kamil Cichostępski *** , **Jerzy Dec *** and **Anna Kwietniak ***

Faculty of Geology, Geophysics and Environmental Protection, Department of Geophysics,
AGH University of Science and Technology, Mickiewicza 30, 30059 Kraków, Poland
* Correspondence: kcichy@agh.edu.pl (K.C.); geodec@agh.edu.pl (J.D.); anna.kwietniak@agh.edu.pl (A.K.)

**Abstract:** In this article, we present a high-resolution shallow seismic surveying method for imaging the inner structure of the Miocene evaporitic formation, where sulfur ore occurs. The survey was completed in the northern part of the Carpathian Foredeep (SE Poland) where sulfur deposits occur up to a depth of ca. 260 m. In this region, the sulfur ore is strata-bound and exists within a carbonate interval of a thickness of approximately 28 m. The average sulfur content reaches up to 30%. Five seismic profiles were acquired with a total length of 2450 m. The acquisition was designed to obtain high-resolution, long offsets and a satisfactory signal-to-noise ratio. In the field, we used 48 channels and variable end-on roll-along spread that allowed us to record offsets of up to 375 m. Data processing was aimed at preserving relative amplitudes (known as RAP, relative amplitude preservation processing), an approach that is necessary for seismic inversion application. With the utilization of well log data and results of simultaneous inversion, we were able to calculate the elastic properties of the deposit to evaluate sulfur ore content and changes in lithology. The sulfur content is strongly dependent on the carbonate reservoir's porosity. To evaluate porosity changes and associated sulfur content, a simultaneous inversion procedure was used. This is a pioneering approach in which we applied pre-stack inversion methods to shallow carbonate sediments.

**Keywords:** sulfur ore deposit; simultaneous inversion; high-resolution shallow reflection seismic; acquisition design; relative amplitude processing

## 1. Introduction

Poland's deposits of native sulfur are among the most abundant in the world [1–3]. They occur within the distal part of the Carpathian Fordeep (southern Poland) at depths of about 300 meters. They are associated with the Miocene evaporitic formation and are strata-bound by the limestone and marly-limestones and marls of various clay additives [4–6]. Sulfur ore occurs in small caverns and fractures; its content averages 25–30% but can also reach as high as 70%. In 2017, the documented resources of sulfur ore were estimated to be 503.85 million tons [7]. Currently, out of 16 known deposits, only one is in production—the Osiek exploration site. The net annual yield of the Osiek exploration site is approximately 600 thousand tons of sulfur. Due to an increasing demand for sulfur [8], the Basznia exploration site will be reactivated; its resources are estimated to be 3.6 million tons of sulfur. Due to its high quality, the sulfur ore obtained will be used in pharmaceuticals and cosmetics. The seismic method was used for recognition and preliminary assessment of the Basznia sulfur deposit.

Seismic imaging of carbonate hydrocarbon deposits focuses on changes of porosity, facial and seismic stratigraphy analysis. In the case of carbonate sulfur deposits, seismic interpretation is mainly focused on porosity changes at the top of the layer and within the carbonate interval. An increase in porosity results in a drop of the amplitude in the wave reflected from the top of the carbonate sediments, which in some cases may result in a dim spot or even a phase reversal (e.g., [9]). To seismically detect

the sulfur ore we should investigate relationships between the amplitude of the reflected wave and porosity, as well as the percentage of sulfur within the cap rock. To quantify the reserve, we have to understand its elastic parameters, lithology and geometric distribution. Yet, the reflectivity seismic section only provides qualitative information and cannot be used for quantitative petrophysical parameter determination. For this, a seismic inversion is needed. This is a technique where the original seismic reflection data are converted into an impedance domain distribution, which is more suitable for reservoir characterization than the raw amplitudes. The strong relationship between the porosity and acoustic impedance in carbonate rocks is the basis for inversion procedures [10–13].

Seismic reservoir characterization for carbonate environments is often complicated and inversion procedure results are not as unequivocal as for clastic environments. Acoustic inversion performed in the zero-offset domain is not sufficient for distinguishing between porous limestone and mudstone [13]. For carbonate environments, the offset-variable analysis is currently gaining momentum. A few authors have presented their results obtained through simultaneous inversion for the determination of porosity and pore infilling in carbonate rocks (e.g., [14–16]). The results of simultaneous inversion, which is performed on offset pre-stack gathers, additionally enable an estimation of the elastic moduli of the rock: bulk modulus ($\kappa$), shear modulus ($\mu$), Young's modulus ($E$) and the Lamè constant ($\lambda$). With these parameters, it is possible to determine the mechanical properties of the exploration site and carry out lithofacie analysis that enables more accurate reservoir estimations. Smith [17] indicated that for carbonate rocks with an increase in porosity, both bulk density and Young's moduli decrease; Ishiyama et al. [16] used shear modulus and the Lamè constant for the evaluation of porosity.

The basic problem in seismic imaging of carbonate intervals is that they are characterized by high heterogeneity in horizontal and vertical directions and, at the same time, by very high velocities. Long wavelengths associated with these velocities are usually unable to map their inner morphology; high velocities in carbonates generally mean that seismic resolution is low (e.g., [18,19]). Another complication within the study area is the rather low thickness of the carbonate sequence—usually between 25 and 30 m. For this reason, for seismic imaging of such a specific case, the acquisition design and processing steps must be adjusted to obtain a high-resolution seismic image.

In this paper, we present the results of high-resolution shallow reflection seismic surveying aimed at imaging the carbonate sulfur deposit "Basznia". According to our knowledge, there has been no previous application of pre-stack inversion for evaluation of sulfur content within a carbonate reservoir. We designed and performed seismic acquisition and processing that allowed us to obtain relative amplitude preserved data which are necessary for reservoir characterization. With the utilization of well logs and simultaneous inversion processed on pre-stack gathers, we obtained P- and S-impedance volumes which were used for determination of shear modulus and the Lamè constant. With these petrophysical parameters, we attempted to evaluate the sulfur deposit to indicate the most favorable drilling areas. Due to the lack of S-wave measurements in the study area, we utilized shear waves that are necessary for the inversion process, through linear relationships with P-wave velocities. Therefore, our result was an approximate assessment of the sulfur deposit. The results of simultaneous inversion are presented by an arbitrarily chosen seismic profile that intersects with the recently drilled well for which we have well log information. With this information, we were able to link the obtained elastic field and petrophysical parameters accurately. With the calibrated results we were able to map the distribution of the elastic parameters away from the well and indicate preferable exploitation sites.

## 2. Study Area and Geological Setting

The study area is located in SE Poland, between the city of Lubaczów and the Ukrainian border (Figure 1a). Native sulfur ore deposits, "Basznia", are striated in shape and of the length of approximately 12 km laying in a NW–SE direction along the distal section of the Carpathian Foredeep (Figure 1c). The Carpathian Foredeep is a flexural basin that contains the youngest, outermost and largely undeformed fill that resulted from the northward thrusting of the Carpathian accretionary prism onto the West and East European platforms in Europe. The fill consists chiefly of Miocene

siliciclastic sediments, up to 3800 m thick, onlapping to the north the Precambrian crystalline basement and its Palaeozoic–Mesozoic sedimentary cover. Southward, the Miocene fill extends more than 50 km below the thrust stack and consists of flysh rocks [20]. The base-of-Miocene basement within the study area consists of Proterozoic, Lower Cambrian, Lower Paleozoic, Jurassic and Upper Cretaceous deposits (Figure 1b). The Miocene succession begins with Lower and Middle Badenian mudstones and sandstones that lie unconformably on Mezozoic strata built-up by Upper Cretaceous carbonates. In the upper profile lies an evaporitic formation composed mostly of sulfurized carbonates, gypsum and marls of the Upper Badenian age. Outlaying the distal part of the Carpathian Basin, an interval occurs, built mainly by anhydrates and gypsums. It is an excellent stratigraphic and seismic marker because, most commonly, it consists of several meters of anhydrites [21]. The evaporitic formation was developed during the Badenian salinity crisis in the northern Central Paratethys [20]. The evaporitic unit is followed upwards by Sarmatian sandstone and mudstone [20,22], which form the rest of the basin infill. The geological profile ends with post-glacial Quaternary strata with a thickness of few meters [23]. The thickness of a clastic series that builds the overburden is variable and reaches 230–370 m. In the survey area, the mineralized limestone lay at a depth of 250 m, and their thicknesses reached up to 25 m.

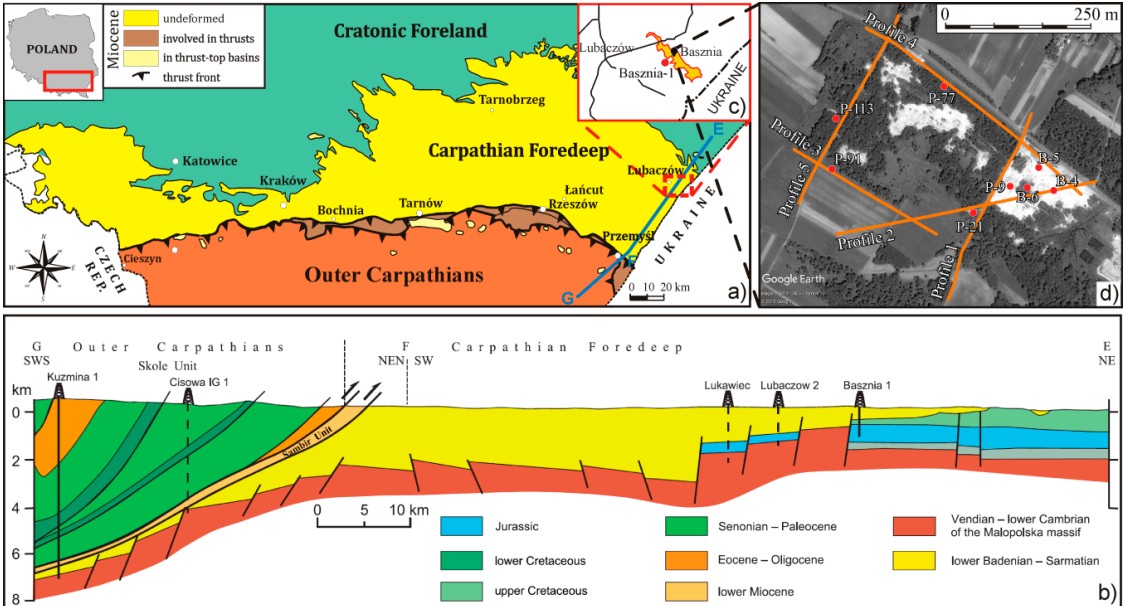

**Figure 1.** (**a**) Location of the study area (dashed rectangle) in the Polish Carpathian Foredeep; (**b**) simplified geological cross-section G–F–E (shown in (a) as a blue line) across study area (modified after [20]); (**c**) inset, shows the Basznia sulfur deposit; (**d**) location of seismic profiles (orange lines) and wellbores (red circles), source of map: Google Earth.

The scope of the survey was to determine the morphology of the top of the evaporitic sequence and perform a preliminary reservoir characterization for the location and design of future exploration wells. The survey site and future exploration area lie within the old sulfur mine that was shut down in 1993 due to the global recession in the sulfur market. The new sulfur mine, Basznia II, will be the second sulfur mine in which the Frash method will be used. The Frash process involves forcing super-heated water into a sulfur deposit, dissolving sulfur and returning it to the surface by way of compressed air. The first sulfur mine using the Frash method was the Osiek sulfur mine located to the west of the survey area. The authors conducted seismic monitoring of the reservoir exploration in the Osiek sulfur mine [24–27].

## 3. Seismic Data Acquisition

The seismic acquisition was completed in January 2018 in the area of the Basznia sulfur deposit. Over three days, we made five seismic 2D reflection profiles with a total length of 2450 m (Figure 1d). For all profiles, we used the same acquisition scheme. The only variables were in offset range and profile length, which were determined by terrain conditions and mine infrastructure. We will describe the acquisition parameters from profile 2 (the location of this profile is shown in Figure 1d).

Seismic data were collected using a 48-channel recording system (Geometrics Geode) with 100 Hz vertical geophones placed 5 m apart. We used a variable end-on roll-along spread with a Gisco ESS-500 Turbo seismic source (accelerated weight drop of 227 kg—impact velocity of 3 m/s with an energy of 1022 J). The shot interval was 10 m. Due to frozen soil, to obtain a satisfactory signal-to-noise ratio at each shot position three weight drops were usually performed. In places where the ground was softer or covered by debris, we performed up to five hits. Subsequently, the corresponding records were vertically stacked. The initial distance between the shot position and the first receiver was equal to 70 m. When the thirteenth receiver was reached, the spread was moved. After the last deployment, when we had reached the thirteen receivers, the source was moved to the thirty-sixth receiver, and shots were performed to achieve a maximum offset of 140 m from the last geophone. The designed geometry leads to an average fold of about 15, with a common midpoint (CMP) spacing of 2.5 m. Table 1 summarizes the main acquisition parameters used to acquire data along profile 2, and Figure 2 presents its field acquisition scheme with fold distribution. Figure 3 shows one of the acquired raw shot gathers. Obtained seismic data had good quality overall, with clear reflections that were observed down to ca. 600 m. Furthermore, with the end-on spread, we recorded larger offsets which is essential for pre-stack inversion.

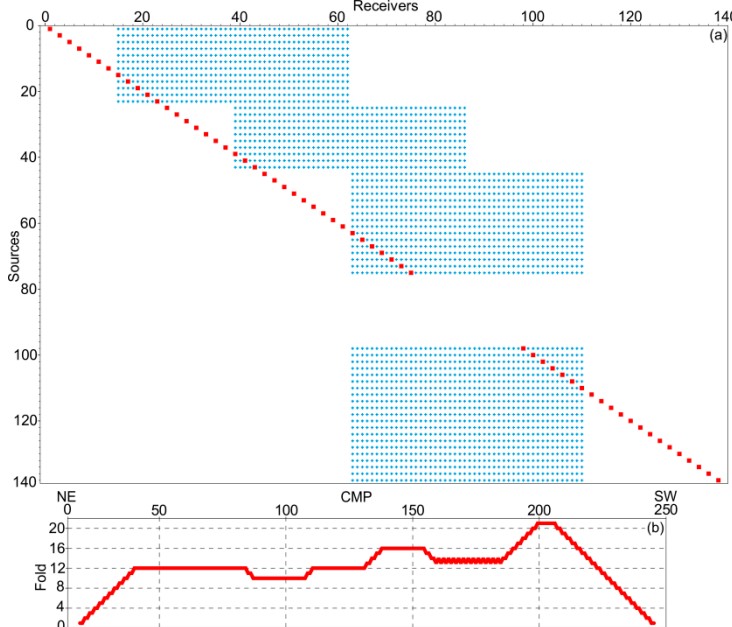

**Figure 2.** Field acquisition scheme of seismic reflection profile 2. (**a**) Stacking chart (blue crosses—receivers, red squares—shots); (**b**) Fold distribution. The total number of shots was 59, with 48 active channels per cable which gave a nominal fold of 15.

**Table 1.** Seismic acquisition parameters.

| Feature | Measurement |
| --- | --- |
| Source Type | ESS-500 Turbo (Accelerated Weight Drop) |
| Recording system | Geometrics Geode |
| Receiver | Single vertical 100 Hz geophone |

**Table 1.** *Cont*.

| Feature | Measurement |
|---|---|
| Geometry | Variable End-on roll-along spread |
| Vertical stacks | 3 times average, max up to 5 |
| Receiver interval | 5 m |
| Shot interval | 10 m |
| CMP interval | 2.5 m |
| #Active channels | 48 |
| Absolute offset range | 0–375 m |
| Average fold | 15 |
| Sampling rate | 0.5 ms |
| Record length | 1 s |

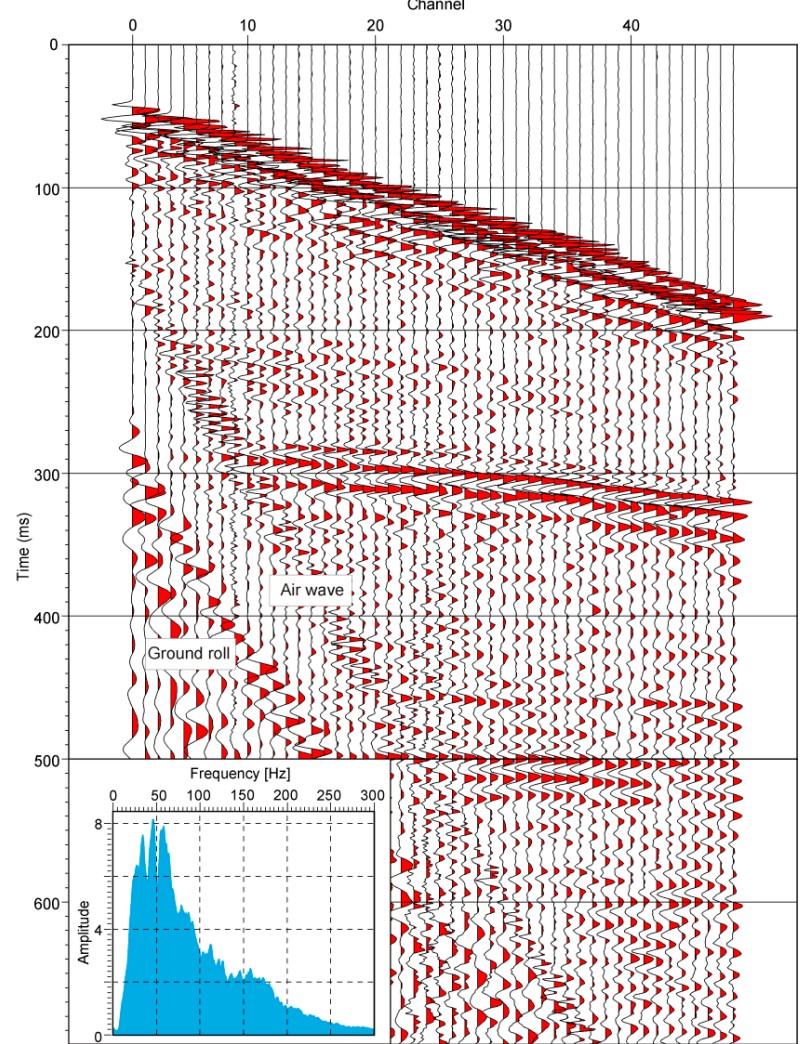

**Figure 3.** Example of shot gather after automatic gain control (for display purposes only) and its amplitude spectrum for the source position 50 m before the first receiver. The coherent noises (ground roll and airwave) are recorded at times below reflections from the top and bottom of the carbonate of the Miocene evaporitic formation that occur at 280 ms.

## 4. Seismic Data Processing

The acquired data were processed with relative amplitude preservation (RAP), which is necessary when using seismic data for seismic reservoir characterization [28–32]. The main goal of the processing was to obtain high quality pre-stack data that would be further used as an input for elastic moduli

extraction through inversion workflow. Also, we obtained a migrated section that would be used for structural interpretation. Data were processed using Vista 2D/3D Seismic Data Processing software. Workflow involved the standard processing path and some advanced processes. The latter including linear and white noise removal with signal preservation, surface-consistent amplitude scaling and deconvolution and offset dependent spherical divergence correction. No invasive methods of signal enchantment were applied, e.g., spectral whitening or bandwidth extension, as these procedures can disturb relative amplitudes. After each procedure, we compared input and output gathers and their amplitude spectra to make certain that the useful signal was not removed and the amplitudes were not damaged. The complete processing sequence is presented in Table 2. Figure 4 shows processed gather from Figure 3 and its corresponding spectrum.

**Table 2.** Seismic data processing sequence.

| Procedure | |
|:---:|:---:|
| Geometry and Trace Edit | |
| Adaptive noise removal with signal preservation (Radial Transform and THOR) | |
| Surface-Consistent Amplitude Scaling (source and receiver) | |
| Surface-Consistent Spike Deconvolution (source and receiver) | |
| Bandpass filtering (40/60–200/250 Hz) | |
| Geometric divergence correction (offset-dependent) | |
| Datum (floating), refraction statics | |
| First break muting | |
| Velocity analysis | |
| NMO | |
| Automatic the residual statics | |
| NMO | |
| Phase rotation | |
| Stack | Gather conditioning |
| Signal enchantment on stack | Simultaneous inversion |
| Post-stack Kirchhoff migration | $\lambda - \rho$ and $\mu - \rho$ extraction |

Preprocessing included geometry application and excluding noisy channels that resulted from poor geophone coupling. For the surface wave removal, a Radial Transform [33,34] was applied. The model of the surface wave was generated within apparent velocities of $-800$ to 800 m/s. An adaptive filter was designed to fit the model traces to the real traces. Then, in the time domain, the filter was applied to the model traces and subtracted from real traces without damaging the signal.

The airwaves and noise burst were effectively removed with the application of THOR procedure [35]. THOR is a threshold noise attenuation/replacement and does not damage the signal. It uses a threshold median replacement in the frequency domain [36,37]. When the trace median calculated in a short fast Fourier transform window exceeds a given threshold, it is treated as noise and replaced. During calculations, we used time window lengths of 60 ms and spatial windows of 13 traces. In the next step, amplitude scaling and deconvolution were performed with the surface-consistent method. In surface-consistent amplitude scaling, we compensate for the effect of variations in amplitude caused by different shot conditions and receiver coupling. For improving vertical resolution and balancing the amplitude spectra, spiking the surface-consistent deconvolution was performed. The length of the operator was set to 80 ms. After deconvolution, we applied a 40/60–200/250 Hz bandpass filter. This high-pass filter was used not only to attenuate high frequency

noise that was provided by the deconvolution process but also to attenuate residues of the surface waves that were not completely removed by the Radial Transform. The subsequent step included geometric divergence compensation (time and offset correction). Then, the seismic data were moved to the floating datum, which was set to smoothed ground elevation. Refraction statics were calculated by the usual approach, we defined a velocity model for calculations from slopes and intercept times of first breaks. Before velocity analysis, we carefully performed top mute to remove direct and head waves with preservation of far offset reflections. Obtained gathers have a spectrum range of about 60–160 Hz (at −3 dB level) and the dominant frequency of ca. 110 Hz (Figure 4). After velocity analysis and residual statics, the gathers were corrected for normal moveout and the phase rotation was applied to obtain the zero-phase signal. Gathers processed in the presented sequence were used as an input for the inversion workflow.

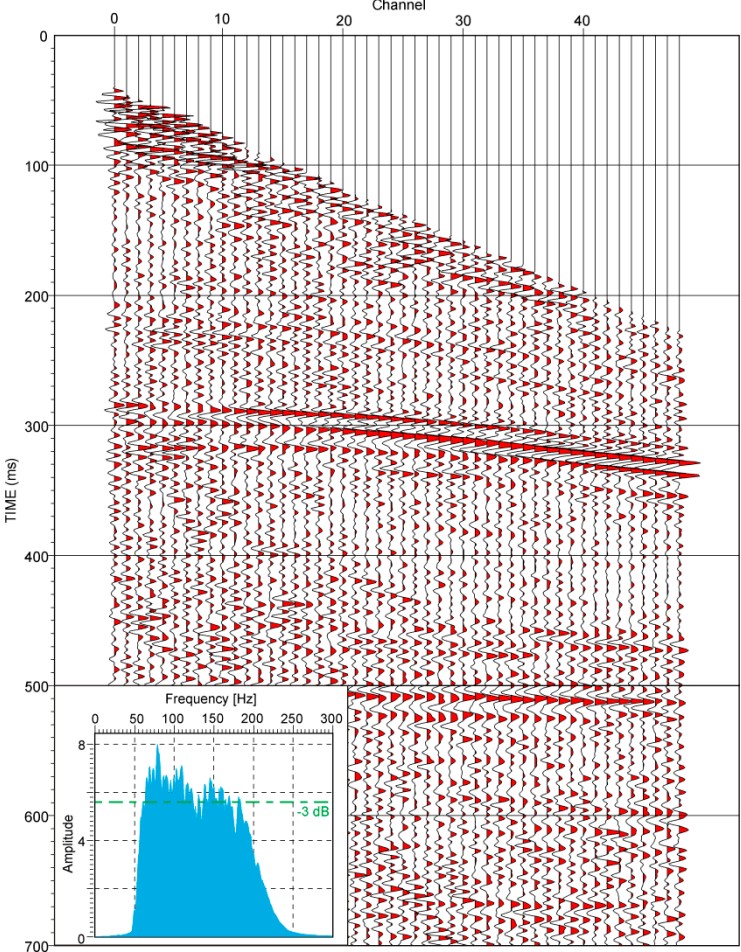

**Figure 4.** Example of shot gather from Figure 3 and its spectrum after noise removal, surface-consistent amplitude scaling and deconvolution, bandpass filtering, geometrical divergence correction, statics and first break muting. Obtained bandwidth is 60–160 Hz (at −3 dB) and the dominant frequency is ca. 110 Hz.

Next, we processed the data further to obtain the migrated section that would be used for structural interpretation. In this additional workflow, the pre-stack gathers after NMO were stacked. For signal enhancement on stacked sections, we utilized an FX Prediction procedure [38]. This procedure works by calculating the Fourier Transform of every trace. Then, the complex frequency samples are multiplexed, and a series of mono-frequency values across time position are obtained. For each mono-frequency series, a two-sided complex Wiener Prediction Filter is calculated. This filter is then applied, and finally, the inverse F-X transform is calculated. The effect of the FX prediction filter

is to smooth data across space. In general, the FX Prediction filter uses short windows. The design window that we used consisted of 20 traces, and filter length consisted of 5 traces. This procedure successfully enabled us to improve the quality of the reflections on the stacked section. Finally stacked seismic data were migrated with the Kirchhoff algorithm using a smoothed version of stacking velocities.

## 5. Rock Properties of Sulfurized Carbonate

Within the survey area, only three recently drilled wells with complete well logs were available: B-4, B-5 and B-6, (Figure 1d). None of these wells had a shear sonic log which is required for determination of the carbonate rock properties. With the application of a solution from Castagna et al. [39] for clastic rocks, and its modification for carbonate rocks given by Li and Downton [14], we performed an S-wave prediction. These formulas' coefficients were modified further by the authors to be more suitable for the non-consolidated rocks of the Carpathian Foredeep. This was done by a comparison with wells that have shear wave logs and are located in the same part of the Carpathian Foredeep as the study area. S-wave velocity for clactic interval was estimated as:

$$Vs = 0.965Vp - 1284 \text{ m/s}, \tag{1}$$

and for the evaporitic sequence as:

$$Vs = 0.4878Vp + 230 \text{ m/s}. \tag{2}$$

These relations are similar to the Osiek sulfur mine where the geological setting is mostly the same and are verified for the reservoir interval [26].

In Figure 5, well information for well B-4 is presented. It can be seen that the distribution of sulfur mineralization and lithology within the evaporitic interval is heterogenous (at depths of 251–278 m). The upper-most reservoir is built by porous, marly limestone of high clay content and low sulfur mineralization (average volumetric sulfur content is 11.6% for average porosity of 25%). The thickness of these sediments is approximately 8 m, and they are underlain by a massive limestone deposit which manifests high sulfur mineralization (average volumetric sulfur content is 41%, for average porosity of 9%).

For the carbonate rocks, we observed a strong linear relationship between sulfur ore mineralization and porosity (Figure 6). This relation was empirically determined with the use of over 50 wells that go through the reservoir interval [25] and demonstrates that with higher sulfur mineralization the porosity steadily decreases.

The velocity model for the reservoir interval is composed of the matrix of carbonate rock and pore-infill (brine and crystalline sulfur). The velocity within the massive carbonate interval reached approximately 4000 m/s. The average P-wave velocity for crystalline sulfur is 2500 m/s [40] while for brine it is 1410 m/s. The sulfur mineralization replaces brine and decreases the porosity which increases both P- and S-wave velocity and, consequently, impedance. This implies that the rock matrix is stiffer for higher sulfur mineralization. With the increase of sulfur mineralization within the carbonate reservoir, values of the elastic moduli ($\lambda$ and $\mu$) also increase. As was noted by Ishiyama et al. [16] change in these parameters is more sensitive to changes in porosity and lithology in comparison to P- and S-impedance alone.

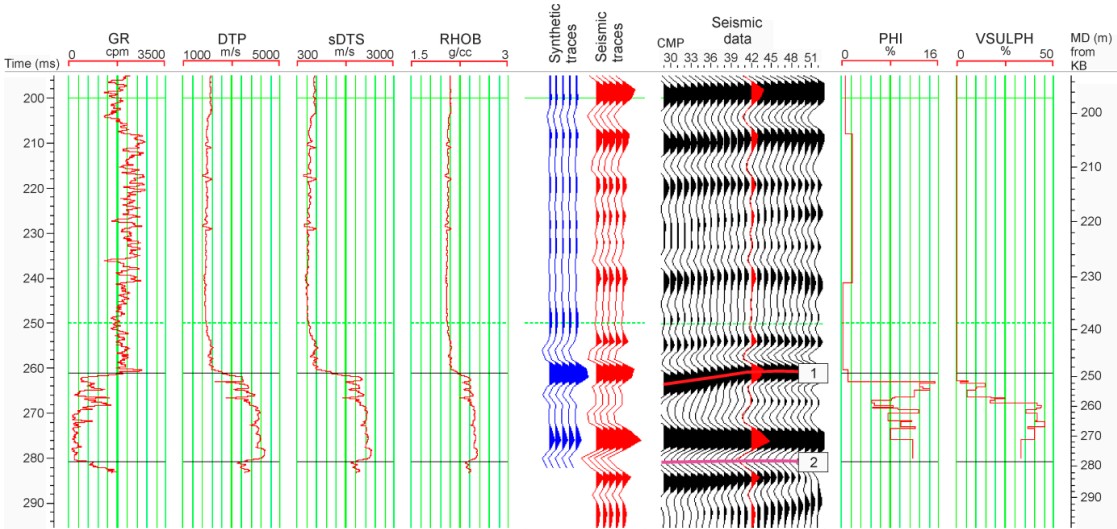

**Figure 5.** Correlation of B-4 well gamma ray (GR), P-wave velocity (DTP), synthetic S-wave velocity (sDTS), density (RHOB), porosity (PHI) and sulfur volume (VSULPH) curves with seismic data. The blue traces represent the synthetics (generated with the full angle wavelet that is shown in Figure 8), whereas the red traces represent the seismic data. The correlation coefficient between the synthetic and red traces for the evaporitic interval of 0.78 suggests satisfactory correlation and enables the mapping of subtle features within the carbonate reservoir due to inversion. 1—top of Miocene evaporitic formation, 2—bottom of Miocene evaporitic formation. The impedance contrast between siliciclastic rocks and the evaporitic formation results in strong seismic reflection.

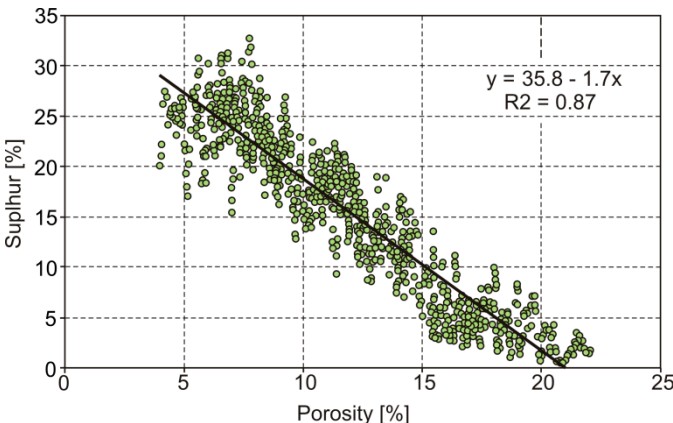

**Figure 6.** The empirical relationship between porosity and sulfur derived for carbonate reservoirs by over 50 wells [25].

## 6. Well-to-Seismic Tie and Seismic Signatures

Profile 2, and its interpretation, is presented in Figure 7. For structural interpretation, data from the most recent wells (B-4, B-5, B-6) and archive profiles (P-9, P-21, P-77, P-91 and P-113) were used (see Figure 1d). All these wells reached a maximum of the bed of the evaporitic sequence. For the identification of deeper layers, we used the data from the old well Basznia-1, which is located near the survey area (see Figure 1c). Due to the high quality of acquired data, the reflections can be unequivocally identified (see Figures 3, 4 and 7). The high positive amplitude event that occurs at the time of ca. 280 ms comes from the Miocene evaporitic formation (it is the first package of high amplitudes within the Miocene sequence), and a subsequent high amplitude positive event that occurs at the time of ca. 500 ms comes from the top of Jurassic rocks. At the time of ca. 450 ms, a negative reflection occurs from the top of the Cenomanian. The positive reflection that appears at the time of

ca. 300 ms comes from the top of the pre-Miocene Cretaceous basement. The reservoir is interbedded by non-consolidated clastic sediments of significantly lower acoustic impedance than the sulfurized carbonates. Such a relation of the petrophysical parameter results in the top of the limestone sequence having very strong positive seismic reflection, and its base very strong negative selection (see Figure 5). In Figure 5, the synthetic seismogram is presented with stacked data. The correlation coefficient between the synthetic and real seismic traces for the zone of interest obtained was 0.78 which suggests that the seismic data give a good image of the petrophysical parameters distribution in the well. This provides a basis for the application of the inversion process for imaging subtle features within the reservoir. The increased sulfur mineralization correlated with the drop in porosity (see Figure 6) and an increase in the acoustic impedance value. The seismic signature of the reservoir series corresponded with the increased value of seismic reflection amplitude.

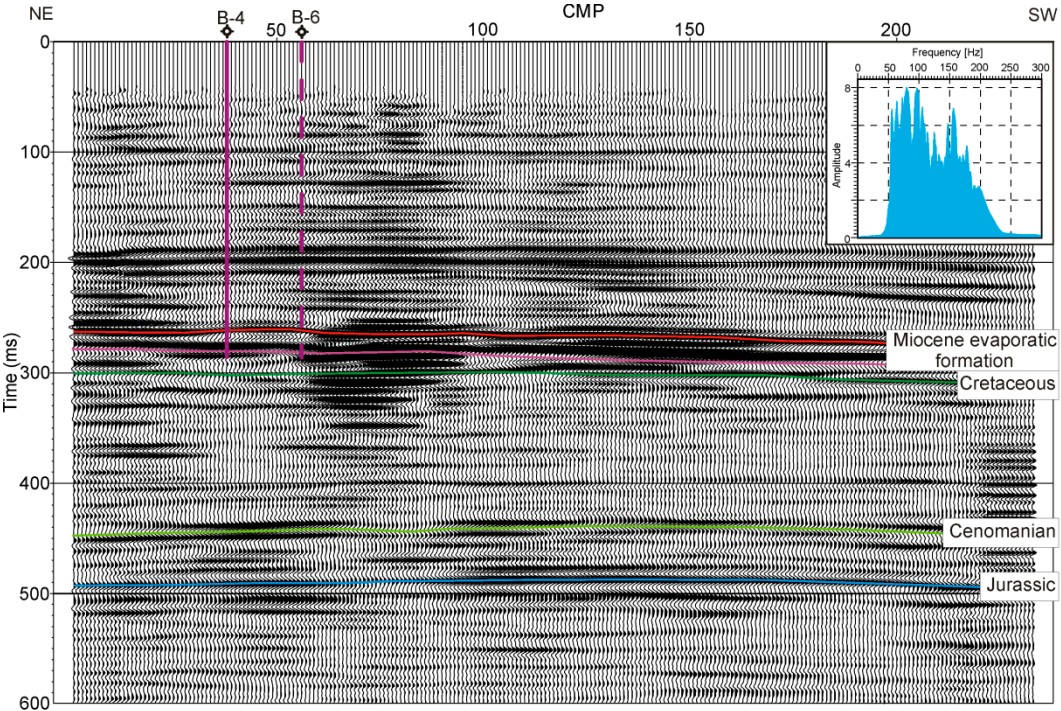

**Figure 7.** Seismic profile 2 and its amplitude spectrum with interpretation and wells position. Well B-6 is projected from 25 m.

By the use of the described processing sequence, the obtained migrated section had a dominant frequency of ca. 110 Hz (see amplitude spectrum of Figure 7). With the average P-wave velocity in carbonate equal to 4000 m/s, the minimum thickness that can be resolved is 9 m ($\lambda/4$) [41]. Because the thickness of carbonate sequence is 28 m there is a possibility for occurrence of a tuning effect that can alter amplitudes. To separate the possible tuning effect from the reservoir characterization, we applied a method of frequency analysis. A method of determining instantaneous frequency can reveal information about the interference of close-lying reflections. It was stated by Puryear and Castagna [42], that when approaching the tuning thickness of a layer instantaneous frequency increases, while instantaneous amplitude increases up to the tuning point and then decreases. The application of instantaneous frequency was also extended in the work of Zeng [43] where the author found another interesting phenomenon for instantaneous frequency—points of rapid decrease of instantaneous frequency indicate layers of decreasing thickness. Such points can determine the spatial distribution of wedges and thin beds. The instantaneous frequency attribute was computed, and its result is presented in Figure 8. The layout of the figure is identical to the seismic section presented in Figure 7. Within the reservoir interval there existed an increase of instantaneous frequency reaching a value of 200 Hz and higher. These layers are indicated by black arrows and are interpreted as thin beds. Additionally,

the rapid changes of the value can be seen—these are indicated by smaller white arrows in Figure 8. These points are interpreted as parts where the thin bed is losing its thickness and creates wedges. Moreover, a zone beneath the reservoir interval, that was characterized by very high amplitudes (see Figure 7) also corresponded to the increase in values of instantaneous frequency. These led to the conclusion that the region (shown by an ellipse in Figure 8) is composed of a finely layered interval that can be classified as thin beds. In this region, the tuning effect can occur.

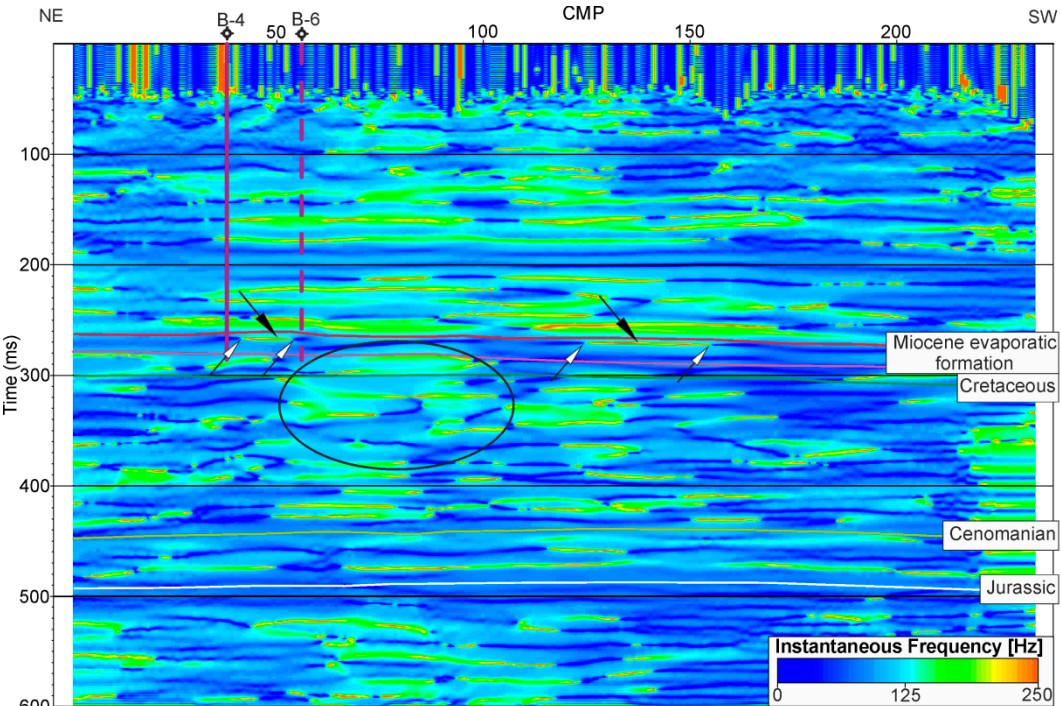

**Figure 8.** Seismic profile 2 in instantaneous frequency attribute with interpretation and wells position. Well B-6 is projected from 25 m. Black arrows indicate thin beds, white arrows indicate places where thin beds lose their thickness. In the area marked by an ellipse, tuning can occur.

## 7. Simultaneous Inversion

Simultaneous inversion [44] is a process where pre-stack gathers are utilized directly to determine the P-wave impedance (compressional impedance)—Zp, S-wave impedance (shear impedance)—Zs and density—ρ. However, pre-stack seismic inversion, like other inversion algorithms, has several limitations. The most important is the absence of low and high frequencies in the seismic spectrum (cf., [45,46]). These absent frequencies can be provided by well log data with the model-based approach. Model-based inversion uses an iterative forward modeling and comparison procedure [46]. In this method, a simple low frequency model of earth's geology is designed through interpolation and extrapolation of well logs (impedance and density logs) along interpreted horizons and then altered until the derived synthetic CMP gathers fit best to the original seismic gathers. For the model creation, we used well B-4. It is the only well that has well log measurements in the overburden. Usually, in an inversion of deep seismic data, impedance logs are filtered (<10 Hz). Because the bandwidth of our shallow seismic is 40/60–200/250 Hz, for the model generation we filtered the P-wave and S-wave impedance logs with a 40/60 Hz high-cut filter.

The input data for inversion are CMP gathers after phase rotation (see Table 2). The obtained zero-phased CMP gathers were carefully conditioned to guarantee that amplitudes were preserved across offset/angle. The idea behind conditioning is to enhance the signal-to-noise ratio. The processes used in conditioning were supergathering (three adjacent gathers were used to create one supergather), random noise attenuation (Radon transform) and trim statics. In Figure 9, angle of incidence

information was extracted from the velocity model generated for the simultaneous inversion process and overlaid on the conditioned offset gathers. Such a plot helps estimation of the range of angles that can be meaningfully used in the inversion process. Because we used variable end-on roll along spread, the acquired maximum offset is 375 m which gives a maximum angle of incidence equal to 34°. With this methodology we lost information below 6° angles, hence we performed simultaneous inversion in this angle range.

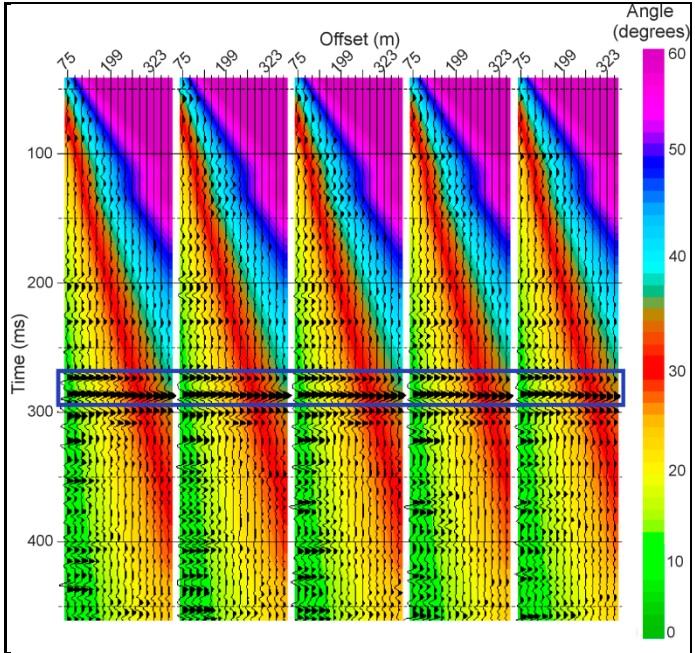

**Figure 9.** The angle of incidence overlaid on conditioned offset gathers. The zone of interest is indicated by the blue rectangle.

In the simultaneous inversion process, multiple partial angle substacks are inverted simultaneously. For each angle stack, a unique wavelet is estimated (Figure 10). The generated low-frequency models for P-impedance, S-impedance and density, wavelets and partial angle stacks were used as an input into the inversion algorithm. The products of the inversion process are P-impedance and S-impedance volumes. The density volume could not be determined by the inversion process because it requires angles beyond 45° that were not recorded during acquisition.

The obtained impedance volumes are shown in Figure 11. The P-wave impedance distribution (Figure 11a) shows the carbonate interval accurately as an increased value within the clastic sequence of the lower impedance. The well log data analysis (see Figure 5) revealed that the increased values of P-wave velocities (and consequently P-wave impedance) might suggest a decrease in porosity that is caused by replacing brine with crystalline sulfur. Nevertheless, the change in value is rather small, and the prospecting sites cannot be simply indicated. Since we obtain S-wave velocities from P-wave velocities through a linear relationship, both logs and impedance volumes show the same pattern (Figures 5 and 11b). Therefore, the obtained impedances are input models for further approximate assessment of the Lamé constant and shear modulus.

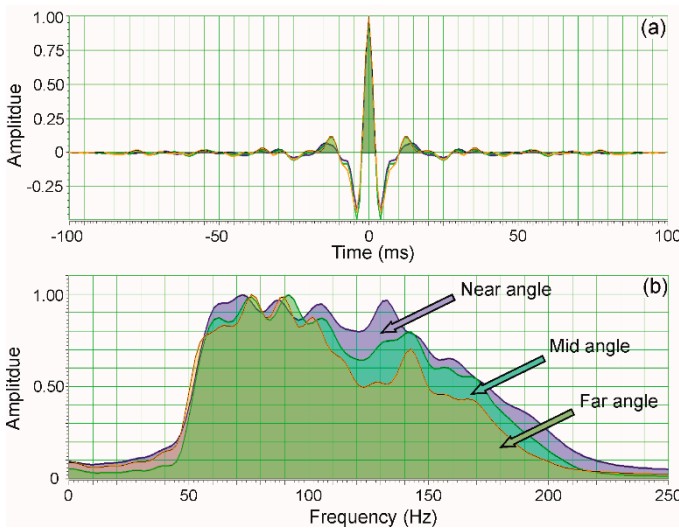

**Figure 10.** (**a**) Zero-phase wavelets estimated for different angle stacks (near, mid and far), which were used in inversion; (**b**) amplitude spectra of estimated angle wavelets. Note the changes in the frequency content of these wavelets.

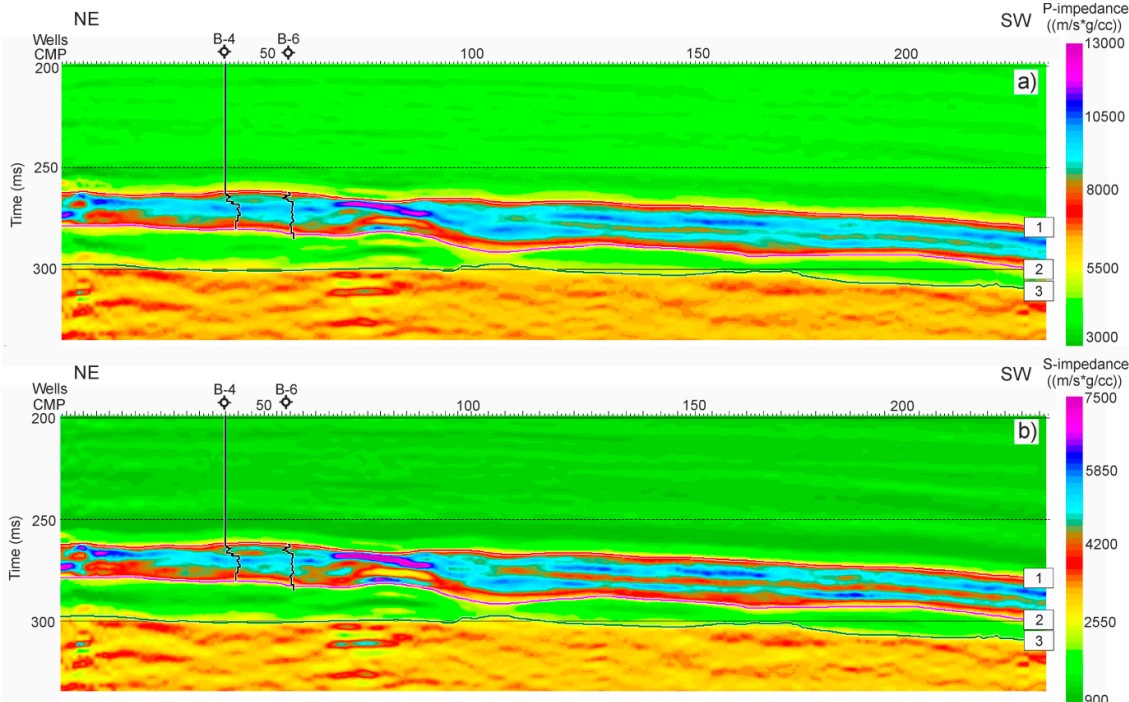

**Figure 11.** Part of profile 2 showing the zone of interest as: (**a**) P-wave impedance, (**b**) S-wave impedance. Well logs indicate sulfur content (%). Well B-6 is projected into the profile with 25 m offset. As can be seen, both impedances give a similar image of sulfur deposit. Horizons 1 and 2 are, respectively, the top and bottom of the evaporitic formation; horizon 3 is the top of the Cretaceous deposits.

As presented by several authors (e.g., [14,16,47]) for reservoir analysis within the carbonate sediments, the crucial parameters are the Lamé constant and shear modulus. These parameters are more sensitive to porosity and lithology changes, thus enabling more precise reasoning for the type of filling in pore spaces than only the P- and S-wave impedances. To compute the values of the elastic

moduli, the LMR method was used [47,48]. This method, however, enables only the computation of λ and μ both multiplied by density. The Lamé constant is computed based on the following relation:

$$\lambda - \rho = Z_P^2 - 2Z_S^2, \tag{3}$$

and shear modulus from:

$$\mu - \rho = Z_S^2. \tag{4}$$

The Lamé constant (λ − ρ) is a measure of the incompressibility of a material. Carbonate rocks with high porosity exhibit lower incompressibility and hence a lower Lamé constant. Therefore, λ − ρ can be used as a porosity indicator. Similarly, shear modulus (μ − ρ) is a measure of the stiffness of a material. Carbonate rocks with higher porosities have a lower stiffness than low-porosity carbonates. Even carbonates with very high porosity have stiffer frames than sandstone or mudstone, and therefore have higher shear modulus values. Thus, μ − ρ can be taken to be a good parameter for lithological discrimination.

Visible in Figure 12a are high values of λ − ρ in a central part of the carbonate sequence suggesting zones of lower porosities—possible sites with higher sulfur mineralization. The μ − ρ value is a measure of stiffness; high values indicate rock of massive structure and low porosity. Through significant variability of μ − ρ value within the reservoir sequence, it is possible to define lateral intervals within the reservoir sequence. The intervals correlate with the zones of different sulfur mineralization and different porosities. The highest values of the μ − ρ attribute correspond to the intervals of massive limestone deposits (Figure 12b). Decreased values of the attribute may indicate increased porosity or the presence of muddy marls. The anomalous zones of inverse attributes (between CMPs 68–95) we associate with the tuning effect of the thin beds sequence mentioned earlier (Figure 8), rather than with real changes in lithology.

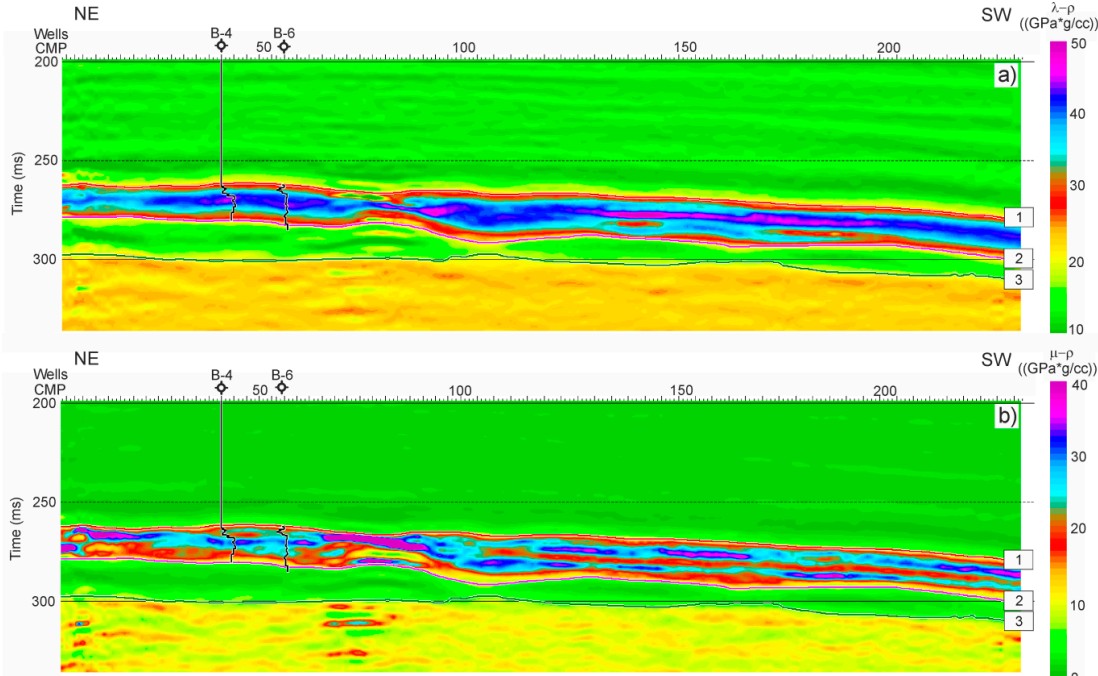

**Figure 12.** Part of the profile 2 showing zone of interest as: (**a**) λ − ρ, (**b**) μ − ρ. Well logs indicate sulfur content (%). Well B-6 is projected into the profile with 25 m offset. High values of λ − ρ may indicate sulfur concentration and the highest values of μ − ρ may indicate a stiffer rock, probably limestone. The image of μ − ρ is more detailed than both volumes of P- and S-Impedances regarding lithology changes. Horizons 1 and 2 are, respectively, the top and bottom of the evaporitic formation; horizon 3 is the top of the Cretaceous deposits.

　　　In Figure 13a, a crossplot of the λ − ρ and μ − ρ values derived from the seismic sections shown in Figure 12 can be seen. With such an approach, more accurate differentiation between different lithologies can be performed with a better understanding of their relations [47,48]. Carbonates of lower porosity lay within the highest values of both inversion attributes. The increase in porosity will result in shifting points towards the lower values of the λ − ρ attribute, and facies changes from limestone to marls and clays will result in drifting of points towards lower values of μ − ρ attributes. The optimal reservoir parameters are marked in the red area in Figure 13a, and the distribution of these parameters is shown in the seismic section (Figure 13b). There exists a direct relationship between the marked area and the increased values of sulfur mineralization indicated by well logs. In Figure 14a,b, the spatial distributions of the averaged values of λ − ρ and μ − ρ for the evaporite sequence are presented. These maps suggest that the Basznia reservoir is relatively homogeneous regarding sulfur mineralization (values of the λ − ρ attribute are between 35–37 GPa·g/cc), but that the facies changes are rather significant (values of μ − ρ attribute fall to the 20–29 interval GPa·g/cc).

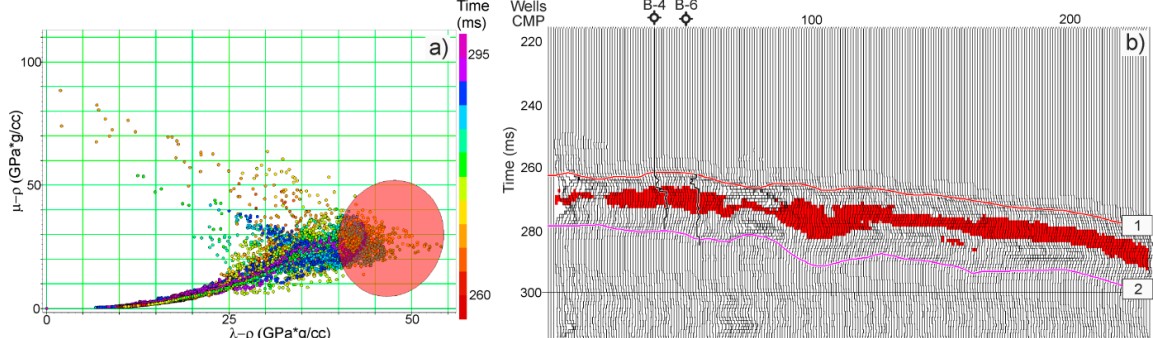

**Figure 13.** (**a**) The crossplot of λ − ρ versus μ − ρ attributes derived from P- and S-impedance values for evaporitic formation; (**b**) Projection of points from a polygon in cross-section display highlights areas where limestone is very stiff and is characterized by high sulfur content. Horizons 1 and 2 are, respectively, the top and bottom of the evaporitic formation. Well logs indicate sulfur content (%).

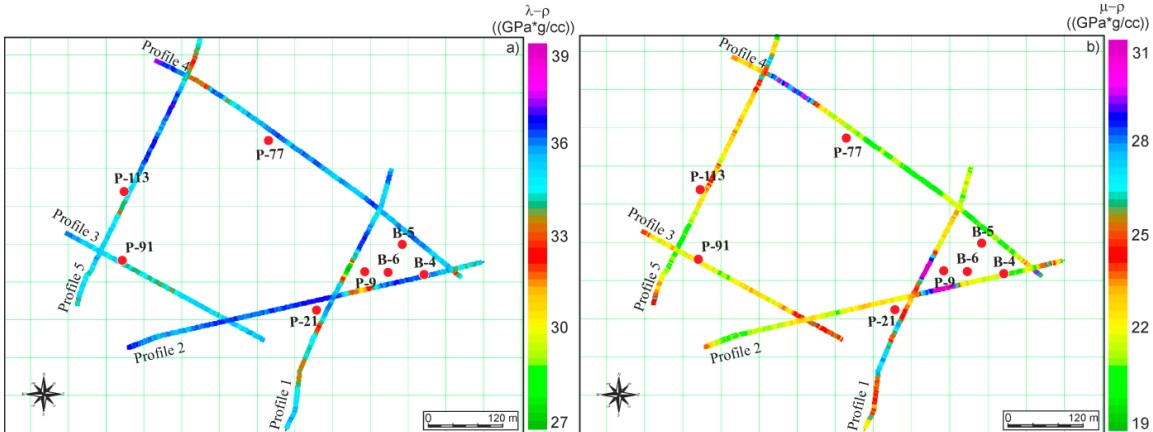

**Figure 14.** A map of root mean square values of (**a**) λ − ρ and (**b**) μ − ρ, along all five seismic profiles derived for the evaporitic formation. The average λ − ρ value is 35–37 GPa·g/cc, which implies homogenous sulfur distribution within the reservoir interval. A wide range of μ − ρ attribute values (20–30 GPa·g/cc) suggests lithofacial variability of the study area.

## 8. Discussion

　　　Shallow seismic reflection imaging requires the special design of acquisition parameters and processing sequence, when it is aimed at petrophysical reasoning due to inversion [31,32]. The main aim of our study was to image a carbonate sulfur deposit that occurs at depths of 260 m and

is 28 m thick. The applied methodology was focused on obtaining high-resolution, long offsets, satisfactory amplitude spectrum, acceptable signal-to-noise ratio and allowance of maintaining relative amplitudes, which are necessary for data inversion. Compared with deep seismic imaging, in shallow seismic reflection imaging, where limited acquisition parameters are used (low numbers of channels), the surface waves are the main issue as they are dominant on shot records [49]. Because we designed and performed seismic acquisition with variable end-on roll along spread, the coherent noises, such as ground roll and airwave, were recorded at times greater than reflections from the top and bottom of the carbonate (Figure 3). Moreover, by omitting the middle part of the profile, we avoided the situation where in the middle spread the surface wave occupies the most of the recorded wavefield. As a result, coherent noise did not interfere with the amplitudes of the target reflections. Hence, the removal of linear noise did not disturb the amplitudes reflected from the geological target [32]. In the acquisition we utilized 100 Hz geophones. According to Knapp and Stepples [50], 100 Hz geophones attenuate unwanted low frequency signal components. Furthermore, 100 Hz geophones allow the obtaining of higher resolution than commonly used low frequency geophones as they perform better in the higher frequency range (75–500 Hz). Nonetheless, the surface wave is strong, which results in increased amplitudes of the frequencies around and below 50 Hz (amplitude spectrum in Figure 3). The influence of the surface wave was reduced during processing by application of the Radial Transform and bandpass filter. With the utilization of deconvolution we restored the energy of higher frequencies which resulted in a spectrum range of approximately 60–160 Hz and a dominant frequency of ca. 110 Hz (Figures 4 and 8). The obtained high quality seismic data had a vertical resolution limit equal to 9 m, which is satisfactory for the reservoir that has a thickness of 28 m.

Because the survey was aimed at the seismic reservoir characterization of a sulfur deposit, the acquired data had to be processed in a RAP manner. RAP-processed sections enable linking of amplitude changes with lateral and vertical changes in the petrophysical properties of a geological medium. The processing criteria for relative amplitude preservation were discussed by, for example, Cambois [30] and Chopra and Castagna [31]. A standard sequence of processing is routinely applied [31,51], although some procedures must be performed with more caution. The most important factor in RAP processing is not to disturb the amplitude relations. Variations in amplitudes should be only caused by changes in the petrophysical parameters between layers [52]. Therefore, for noise removal we used adaptive subtraction algorithms (Radial Transform and THOR procedures). In the adaptive subtraction method, the model traces (noise traces) have different amplitudes, frequency ranges and phases in comparison to the recorded energy (real traces). In order to perform the effective subtraction, the model traces need to be synchronized by amplitudes, time, phase and frequency, with real traces. The adaptive subtraction incorporates the model to match the recorded seismic records to ensure that the subtraction is effective without damaging the signal.

In RAP processing the only amplitude recovery procedures that can be applied are surface-consistent scaling and spherical divergence correction. However, variations in amplitude, phase or frequency partially depend on changes in source and receiver conditions (e.g., the change of the shot position from hard ground to softer). To compensate for these effects, a surface-consistent method was used [53]. According to Ursin [54], gradual loss of amplitudes, which is a result of waveform spreading, is substantial for shallow seismic boundaries and far offsets. Therefore, besides time correction, a correction for offset was also applied. To compensate for near-surface wavelet distortion, the surface-consistent approach in deconvolution was also used. Only such processed volumes can be treated as input data to any reservoir analyses including inversion.

Seismic inversion is a technique where the original seismic reflection data is converted into its impedance distribution [55–58]—A quantitative rock property which is more suitable for reservoir characterization than the raw amplitudes [57,58]. The standard acoustic inversion procedure that is performed on the post-stack data indicates only P-wave impedance distribution for a zero-angle reflection of a seismic wave. In this scenario, information about amplitude changes with offsets/angle that are caused by changes in the Poisson ratio or Vp/Vs ratio is not taken into account. Simultaneous

inversion is performed on pre-stack gathers and enables calculation of P- and S-impedances, as well as density, from the angular dependence of an amplitude reflection, which can be used for further extraction of elastic moduli, such as the Lamè constraint and shear modulus. This allows for better insight into the petrophysical properties of the geological medium [15,16,31,44].

Within the carbonate, with the increase of sulfur mineralization we observed a decrease of porosity. Altogether, with the higher sulfur mineralization the rock matrix was more stiff, and the reservoir was classified as a massive carbonate. In these conditions, such zones will have higher P- and S-wave impedance values, which consequently give higher values of other elastic moduli. For such a petrophysical scenario, the zones of higher sulfur content will give higher seismic reflection coefficients and higher seismic amplitudes.

The analysis of $\lambda - \rho$ and $\mu - \rho$ attributes showed that the reservoir had insignificant mineralization changes but substantial lithofacial diversity. For the reservoir sequence of 28 m, and resolution of 9 m, the horizontal variations of the parameters were more precise than their vertical variations. Unfortunately, only three well logs were available for interpretation. Furthermore, the S-wave logs that were used for the calculation of impedance volumes were estimated through linear relationships from P-wave logs. Hence the mineralization estimation has only a qualitative character and can only be used for preliminary reservoir reasoning. If more precise quantitative analysis is required, then more measured S-wave data from additional wells would be needed. Currently, we are trying to incorporate other seismic attributes to link sulfur mineralization and seismic response in order to find a relation in the case of unmeasured S-wave velocity. Such a method would significantly limit the cost of the analysis.

## 9. Conclusions

In this article, we have presented the application of a high-resolution seismic survey and inversion procedure for imaging the inner variability of an evaporitic sequence. Our main scope was to define the porosity distribution and sulfur mineralization. For these goals, the choice of simultaneous inversion was novel but served the given purpose. Some conclusions are worth noting:

(1) The increase of sulfur mineralization within the carbonate reservoir caused a drop in porosity. With the higher sulfur mineralization the rock matrix was more stiff, and the reservoir was classified as a massive carbonate.

(2) The seismic signature of the sulfur deposit corresponded to an increased value of seismic reflection amplitude.

(3) Extraction and analysis of inversion attributes allowed the interpreter for indicate the most perspective zones within the carbonate sequence. Areas with high sulfur content were characterized by high values of both P- and S-impedances.

(4) The Lamé constant and shear modulus are more sensitive to porosity and lithology changes and enable more precise reasoning for the type of filling in pore spaces than with only P- and S-impedances. Areas with an increased sulfur component had high Lamé constant values. Shear modulus allowed for evaluation of lithofacial changes within the carbonate sequence.

(5) With the use of instantaneous frequency, it was possible to determine zones where the tuning effect may interfere with and blur the inversion results, however, with its prior localization the tuning zone could be treated with less confidence in terms of the reservoir reasoning.

(6) Application of the simultaneous inversion process for near surface data is possible and can be applied to gain insight into the distribution of the petrophysical parameters of shallow geological targets. For such an application, seismic acquisition must be designed to obtain shallow reflections of satisfactory resolution, which typically excludes data that were focused on deep seismic imaging.

**Author Contributions:** Conceptualization, K.C., J.D. and A.K.; Methodology, K.C., J.D. and A.K.; Software, K.C.; Validation, J.D. and A.K.; Formal analysis, K.C., J.D. and A.K.; Investigation, K.C., J.D. and A.K.; Resources, K.C.,

J.D. and A.K.; Data curation, J.D. and K.C.; Writing—original draft preparation, K.C.; Writing—review and editing, A.K., J.D. and K.C.; Supervision, J.D.; Project administration, J.D.; Funding acquisition, J.D.

**Funding:** The research was supported by Polish Sulphur LTD, the owner of the Basznia II sulfur mine and AGH University of Science and Technology in Krakow (Poland), grant number 11.11.140.645.

**Acknowledgments:** We thank the Geophysical Services Company "Geokar-PBG" LTD for providing all the necessary well data. Vista 2D/3D Seismic Data Processing and the Hampson-Russell Suite were provided by Schlumberger and CGG was provided through the University Software Donation Program. We would also like to thank the two anonymous reviewers for providing valuable and insightful feedback on the manuscript.

**Conflicts of Interest:** The authors declare no conflicts of interest.

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
