# Peer review of "Simultaneous Inversion of Shallow Seismic Data for Imaging of Sulfurized Carbonates"

_minerals, doi:10.3390/min9040203_

Round 1
Reviewer 1 Report
Review - Simultaneous inversion of shallow seismic data for imaging sulphurized carbonates
This paper presents a case study of inversion in shallow seismic data that is applied to high quality data, is very carefully done and explained and is definitely worthy of publication with some minor modification of word usages and some expansion of discussion on a couple points.
I have made a number of suggestions for modifying word choices to make the text more consistent with more common English usages.
The several points on which I would like to see more discussion are:
line 71 - I am quite sure that there are some other examples of inversion on shallow seismic data and suggest a more thorough literature search but I would agree that this is likely the first attempt at characterizing sulphur reserves.
Line 232 - requires a bit more explanation. How were the formulas modified and what is the basis for the modification? This is important because the calculation of Vs and subsequent inferences for λ & μ form the basis for the interpretation.
Line 280 - This is a normal value for correlation between real and synthetic seismic data but at 0.65 the correlation is actually quite modest. I think the quality of the correlation is a bit overstated here. My experience with this type of comparison is that the correlation is quite high for major features and significantly lower for less prominent features. So the quality of the correlation is dependent on the suptelty of the feature that is the target for the inversion. Have you determined the correlation for just the zone of interest? That may be more informative than the overall correlation.
Line 346 - I am not sure if you are implying a causal relationship here. I presume that the decrease in porosity is an indication of increased in-fill of the porosity by the precipitated sulphur rather than the other way around.
Line 376 - This of course opens up the possibility that tuning effects could be making significant contributions to amplitude variations. A short discussion of why you make this statement and how that zone differs from others would be beneficial here.

Author Response
Review - Simultaneous inversion of shallow seismic data for imaging sulphurized carbonates
This paper presents a case study of inversion in shallow seismic data that is applied to high quality data, is very carefully done and explained and is definitely worthy of publication with some minor modification of word usages and some expansion of discussion on a couple points.
I have made a number of suggestions for modifying word choices to make the text more consistent with more common English usages.
The several points on which I would like to see more discussion are:
line 71 - I am quite sure that there are some other examples of inversion on shallow seismic data and suggest a more thorough literature search but I would agree that this is likely the first attempt at characterizing sulphur reserves.
Line 232 - requires a bit more explanation. How were the formulas modified and what is the basis for the modification? This is important because the calculation of Vs and subsequent inferences for λ & μ form the basis for the interpretation.
Line 280 - This is a normal value for correlation between real and synthetic seismic data but at 0.65 the correlation is actually quite modest. I think the quality of the correlation is a bit overstated here. My experience with this type of comparison is that the correlation is quite high for major features and significantly lower for less prominent features. So the quality of the correlation is dependent on the suptelty of the feature that is the target for the inversion. Have you determined the correlation for just the zone of interest? That may be more informative than the overall correlation.
Line 346 - I am not sure if you are implying a causal relationship here. I presume that the decrease in porosity is an indication of increased in-fill of the porosity by the precipitated sulphur rather than the other way around.
Line 376 - This of course opens up the possibility that tuning effects could be making significant contributions to amplitude variations. A short discussion of why you make this statement and how that zone differs from others would be beneficial here.
Response to reviewer #1:
Thank you very much for your comments on our paper. We tried our best to improve the manuscript
and made changes according to your valuable comments. We appreciate for your work earnestly and hope that the corrections will meet the approval. Our answers and comments are listed below.
General comment :
I have made a number of suggestions for modifying word choices to make the text more consistent with more common English usages.
Response:
Thank you very much for language corrections and suggestions. We applied almost all of them. We think that the term “seismic reservoir characterisation” is more commonly utilised in seismic imaging than “seismic reserve characterisation”. Also, the article was proof-read by a native speaker. Now the readability should be improved.
Point 1:
line 71 - I am quite sure that there are some other examples of inversion on shallow seismic data and suggest a more thorough literature search but I would agree that this is likely the first attempt at characterizing sulphur reserves.
Response 1:
We did not find any article about the application of simultaneous inversion for shallow deposits, but yes, it does not mean that it does not exist. We deleted this part of the sentence.
Point 2:
Line 232 - requires a bit more explanation. How were the formulas modified and what is the basis for the modification? This is important because the calculation of Vs and subsequent inferences for λ & μ form the basis for the interpretation.
Response 2:
Castagna mudrock equation that we used always need to be conditioned for the local geology that is used for especially when we deal with unconsolidated sediments. In our case, we modified this relationship coefficients’ by comparison with wells that have S-wave measurements and penetrate similar lithology in the same part of Carpathian Foredeep as a study area. We add an explanation to the text.
Point 3:
Line 280 - This is a normal value for correlation between real and synthetic seismic data but at 0.65 the correlation is actually quite modest. I think the quality of the correlation is a bit overstated here. My experience with this type of comparison is that the correlation is quite high for major features and significantly lower for less prominent features. So the quality of the correlation is dependent on the subtlety of the feature that is the target for the inversion. Have you determined the correlation for just the zone of interest? That may be more informative than the overall correlation.
Response 3:
The correlation was determined for the whole interval. For the zone of interest, it is 0.78. We change this in manuscript.
Point 4:
Line 346 - I am not sure if you are implying a causal relationship here. I presume that the decrease in porosity is an indication of increased in-fill of the porosity by the precipitated sulphur rather than the other way around.
Response 4:
We changed this sentence to be more precise.
Point 5:
Line 376 - This of course opens up the possibility that tuning effects could be making significant contributions to amplitude variations. A short discussion of why you make this statement and how that zone differs from others would be beneficial here.
Response 5:
Yes, the tuning effect always has a contribution to amplitude variations and hence affect inversion. We added a new figure (8) and discussed our statements.
Point 6:
Line 289 - I don't doubt this but a spectrum post-processing would be interesting documentation.
Response 6:
We add a post-processing amplitude spectra for comparison – see figure 7. The obtained bandwidth is around 60 - 160 Hz (at -3 dB). The dominant frequency is around 110 Hz (we miscalculated the dominant frequency, now we change this in the text) which gives a resolution of 9 m.
Reviewer 2 Report
Dear authors,
I congratulate you on conducting an interesting study showing relevance to broader audience. The study itself has numerous good elements, however, needs more work before it is ready for publication. I attach in an annotated version of your text with questions that need addressing. In addition to these, here is the summary of main things that need modifications:
Introduction needs to be rewritten to have a logical tread and clean line of thoughts. At this stage, it is a lot of random things thrown in one place without logical connection. Only start a new paragraph when it is a new line of thoughts, not connected to previous part.
There are too many speculations in the methodological and results part that are misplaced. Stick to the facts in these sections and keep speculations for discussion.
Conclusion and results need to be separated, all speculations moved into discussion and the discussion itself strengthen by further supporting references.
Conclusion needs to be rewritten to show clearly and concisely what was done and the main findings of the study.
Kind regards

Author Response
Dear authors,
I congratulate you on conducting an interesting study showing relevance to broader audience. The study itself has numerous good elements, however, needs more work before it is ready for publication. I attach in an annotated version of your text with questions that need addressing. In addition to these, here is the summary of main things that need modifications:
Introduction needs to be rewritten to have a logical tread and clean line of thoughts. At this stage, it is a lot of random things thrown in one place without logical connection. Only start a new paragraph when it is a new line of thoughts, not connected to previous part.
There are too many speculations in the methodological and results part that are misplaced. Stick to the facts in these sections and keep speculations for discussion.
Conclusion and results need to be separated, all speculations moved into discussion and the discussion itself strengthen by further supporting references.
Conclusion needs to be rewritten to show clearly and concisely what was done and the main findings of the study.
Kind regards
Response to reviewer #2:
Thank you very much for your comments on our paper. We tried our best to improve the manuscript
and made changes according to your valuable comments. We appreciate for your work earnestly and hope that the corrections will meet the approval. Our answers and comments are listed below.
General comment:
Introduction needs to be rewritten to have a logical tread and clean line of thoughts. At this stage, it is a lot of random things thrown in one place without logical connection. Only start a new paragraph when it is a new line of thoughts, not connected to previous part.
There are too many speculations in the methodological and results part that are misplaced. Stick to the facts in these sections and keep speculations for discussion.
Conclusion and results need to be separated, all speculations moved into discussion and the discussion itself strengthen by further supporting references.
Conclusion needs to be rewritten to show clearly and concisely what was done and the main findings of the study.
Response:
We rearranged, moved and rewrite some parts of the manuscript as suggested. Now the text should be more readable and logical. Also, the article was proof-read by a native speaker to enhance language.
Point 1:
Line 29 - This is statement that either needs to be supported by a peer-reviewed reference or down toned.
Response 1:
We support our statement with three references. Native sulphur deposits that occur in Poland are among the most abundant in the world.
Point 2:
Line 40 - This is only partly true. The interpretation does not necessarily needs to be focused on porosity changes.
Response 2:
Yes, we agree that the seismic interpretation of carbonates does not necessarily need to be focused on porosity changes. In the case of seismic hydrocarbon exploration, it also focused on facies and seismic stratigraphy analysis. When we evaluate sulphur deposits porosity is the most direct indicator. We change this sentence accordingly.
Point 3:
Line 84 - Entire section discusses the same story, hence starting the sentences with new rows are not needed. It is the same line of thoughts
Response 3:
We rearrange this section as suggested.
Point 4:
Line 104 – Overburden?
Response 4:
Yes, we mean overburden.
Point 5:
Line 124 - Your amplitude spectra are not broad, judging by the one shown in Figure 3. Avoid speculations, just state the facts about what was done.
Response 5:
The spectrum that is shown in figure 3 is raw spectra – it is contaminated with surface wave. This is why most energy is located below 50 Hz, and higher frequencies are not that much visible. We add a new figure (Figure 4) presenting shot gather after processing with its amplitude spectra. We obtain spectra with bandwidth 60 ~ 160 Hz (at -3 dB), which we find quite broad. Also, we add spectra of the migrated section for comparison. Nevertheless, we change broad to satisfactory.
Point 6:
Line 130 - This is purely speculative and based on a single paper more than 30 years old! Leave speculations about why you should use the 100 Hz geophones in the discussion part. Just state the facts here about what was used. The reference should be 26, not 25
Response 6:
Yes, this is rather old paper but is broadly cited in the literature. It gives foundations for shallow reflection seismic and shows a comparison of different equipment and its influence on seismic signal. We changed the citation number and moved this section to the discussion part.
Point 7:
Line 135 - Frozen soil provides good coupling by itself. It is enough to say: "Three repeated hits per shot location were and later vertically stacked to increase S/N ratio"
Response 7:
Generally, we agreed, but in places where the ground was softer or covered by debris we performed up to five hits. We have included this statement in table 1, but did not write in manuscript. We now included this information in the text.
Point 8:
Line 159 - What is this interpretation based on? If it is based on a log data, this needs to be supported by a reference or a well log. There is no need to interpret the origin of reflections seen in this section. Leave that for Results part.
Response 8:
The interpretation is based on lithology from old well Basznia-1 which located in the vicinity of the study area (see figure 1, we add the location of well Basznia-1 in the (c) inset). We moved the interpretation to results as suggested.
Point 9:
Line 165 - This part belongs to discussion, not here
Response 9:
We moved this section to the discussion as suggested.
Point 10:
Line 176 - Which software/code was used for processing?
Response 10:
We used Schlumberger’s Vista 2D/3D Seismic Data Processing
Point 11:
Line 184 - There is a lot of steps here that can cause wavelet distortion and amplitude changes. The raw data example gather needs to complemented by the data after processing. They need to come side-by-side to compare the traces and this needs to be accompanied by the corresponding amplitude spectra.
Response 11:
It is true that some steps can distort amplitudes. They require more attention in the case of RAP. We think that we described all procedures that were applied and how we preserve the distortions of amplitudes. We added the final gather with its spectra to compare our processing with raw gather (Figure 4).
Point 12:
Table 2 – Noise removal with signal preservation. What are the details of this step?
Response 12:
This step included radial transform for surface wave removal and THOR for airwaves and noise burst removal. Both procedures were applied with caution to preserve amplitude distortion. We add information about the adaptive filter in Radial transform.
Point 13:
Line 198 – seismogeological layers - ?
Response 13:
It is a slip of the tongue. We mean layers. Changed.
Point 14:
Line 199 - You mentioned earlier frozen soil during acquisition. This should provide relatively uniform coupling and therefore make the surface consistent manipulation an unnecessary step.
Response 14:
We extend information about the acquisition (please see point 7). Because of the non-uniform acquisition conditions, we find sc algorithms necessary.
Point 15:
Line 214 - Surgical mute?
Response 15:
We performed top mute. This sentence was changed accordingly.
Point 16:
Line 211 - It is too many processing steps involved to call this RAP processing. Effect of processing steps needs to be shown on raw versus processed gathers to judge properly!
Response 16:
The workflow for obtaining RAP sections that we used was initially developed by Resnick (1993), discussed by Cambois (2001) and further modified by Chopra et al. (2014). We also find it optimised for shallow seismic (Cichostępski et al., 2019). It gives overall good results and preserves signal in general. We did not perform any unnecessary steps (agc, whitening etc.), we kept it at a minimum to have good quality data and preserved signal. However, none of the processing sequences can preserve the signal completely, unless we use only static correction, nmo and stacking. Almost always is necessary to include steps for enchanting S/N ratio. We included final gather with its spectra for comparison with raw gather to judge our processing flow (new Figure 4).
Point 17:
Line 235 - Some additional peer-reviewed refulgences would help to strengthen the validity of the relations
Response 17:
Only in one well in Osiek sulphur deposit the shear wave logging was performed, and this was described in the cited reference. Obtained relations in this well is very similar to what we have obtained. There are no other references on this because we are only one group in Poland that is taking geophysical investigation for sulfurized carbonates. However, we extended the description of how we obtain those relationships.
Point 18:
Line 260 - Unclear sentence. Please reformulate
Response 18:
The sentence was reformulated. Now it should be more understandable.
Point 19:
Line 267 – discussed?
Response 19:
We change to “noticed.”
Point 20:
Line 288 - This is incorrect. The dominant frequency of your raw data, as shown in Figure 3 is around 55 Hz! Even after all processing steps done, it is highly unlikely that it will become 125 Hz. Again, you need to show raw versus processed example shot gather and their amplitude spectra to judge your processing flow!
You are here calling on Figure 8 before introducing the Figure 7. You can say just:"Shown later in the text or shown in one of the following figures."
Response 20:
Figure 3 presents the raw shot and its spectra (please see point 5). We added spectrum of processed shot gather for comparison (Figure 4) and the final migrated section (Figure 7). After we remove surface wave and perform deconvolution, we restore the energy of high frequencies which resulted in spectra of 60 ~ 160 Hz bandwidth (at -3 dB).
Yes, we miscalculated the dominant frequency of our data. As you can see on amplitude spectrum (Fig 4 and 7) and in new figure presenting instantaneous frequency (new Figure 8) obtained dominant frequency within the reservoir is around 110 Hz, which means that the minimum thickness that we can resolve is 9 meters. We changed this accordingly.
Yes, it is rather inappropriate way to present figure 8 before figure 7. We added amplitude spectra of the migrated section, and we now call it here.
Point 21:
Line 288-294 - This entire part is incorrect and needs to be rewritten with respect to what is actually in the data!
Response 21:
We rearranged this part and removed information about 100Hz geophones because it was misleading.
Point 22:
Line 341-346 - This comparison is meaningless. Since you have used P-wave sonic log, together with modified Castagna's relation to generate the S-wave sonic log, it is very likely they will show same structure. Do not discuss how good they look like, just say that they are the input models used to obtain a rough assessment of Lame's parameters. Earth is not a homogeneous and isotropic media where an assumed relationship of P- and S-wave velocities represent ground truth. The shear modulus you obtain is a therefore a rough assessment, please be clear on this in the text.
Response 22:
Yes, we obtain S-wave through a linear relationship with P-wave, that is way logs and volumes of P- and S- impedance look similar. We add the clarification to the text as suggested.
Point 23:
Line 381-385 - This is all an approximate assessment. You have to state it clearly in the text.
Response 23:
We added clarification (please see point 22).
Point 24:
Line 411 - This part needs to be split in two. 8. Discussion and 9. Conclusion. All speculations, claims why the acquisition and processing selected was chosen, along with further references supporting your results and the methods chosen for inversion should come here. You need to discuss what was done and what makes the selected steps correct.
The conclusion itself is a short summary of the initial hypotheses attacked. What was the hypothesis, how you have attacked it (one or two sentence) and what were your key findings of this study. Please rewrite this with respect to the study itself.
Response 24:
We rearrange chapter 8 and spit this into two separate chapters. Also, we rewrite the text in those parts as suggested.
Point 25:
Line 422 - This is not true, please rephrase with respect to what is seen in your data.
Response 25:
The obtained resolution is about 10 meters as we mentioned earlier in comments. But we made a mistake here, the thickness of carbonate formation is 28 m, not 25 m. We change that in the text.
Point 26:
Line 428 - This is an honest approach that should be kept throughout the paper.
Response 26:
Yes, we added clarification and tried to keep this approach thought the paper.
Point 27:
Line 437 - This is a redundant statement and rather unsupported by other studies. Please remove or rephrase
Response 27:
We removed the statement as suggested.
Round 2
Reviewer 2 Report
Thank you for all your corrections.
Please read trough the manuscript one more time before final publication to avoid any possible text mistakes.
Good luck with your future work